# Health Potential of Clery Strawberries: Enzymatic Inhibition and Anti-*Candida* Activity Evaluation

**DOI:** 10.3390/molecules26061731

**Published:** 2021-03-19

**Authors:** Francesco Cairone, Giovanna Simonetti, Anastasia Orekhova, Maria Antonietta Casadei, Gokhan Zengin, Stefania Cesa

**Affiliations:** 1Dipartimento di Chimica e Tecnologie del Farmaco, Università degli Studi di Roma “La Sapienza”, Piazzale Aldo Moro 5, 00185 Roma, Italy; francesco.cairone@uniroma1.it (F.C.); mariaantonietta.casadei@uniroma1.it (M.A.C.); 2Dipartimento di Biologia Ambientale, Università degli Studi di Roma “La Sapienza”, P.le Aldo Moro 5, 00185 Rome, Italy; giovanna.simonetti@uniroma1.it; 3Dipartimento di Sanità Pubblica e Malattie Infettive, Università degli Studi di Roma “La Sapienza”, P.le Aldo Moro 5, 00185 Rome, Italy; anastasia.orekhova@uniroma1.it; 4Department of Biology, Science Faculty, Selcuk University, 42130 Konya, Turkey; gokhanzengin@selcuk.edu.tr

**Keywords:** *Fragaria* spp., food processing, polyphenols, bioactivity, enzymatic modulation, anti-*Candida* activity, *Galleria mellonella* model

## Abstract

Strawberries, belonging to cultivar Clery (*Fragaria* × *ananassa* Duchesne ex Weston) and to a graft obtained by crossing Clery and *Fragaria vesca* L., were chosen for a study on their health potential, with regard to the prevention of chronic and degenerative diseases. Selected samples, coming from fresh and defrosted berries, submitted to different homogenization techniques combined with thermal and microwave treatments, had been previously analyzed in their polyphenolic content and antioxidant capacity. In the present work, these homogenates were evaluated in relation to their enzymatic inhibition activity towards acetylcholinesterase and butyrylcholinesterase, α-amylase, α-glucosidase and tyrosinase. All these enzymes, involved in the onset of diabetes, and neurodegenerative and other chronic diseases, were modulated by the tested samples. The inhibitory effect on tyrosinase and cholinesterase was the most valuable. Antifungal activity against *Candida albicans*, recently shown to play a crucial role in human gut diseases as well as diabetes, rheumatoid arthritis and Alzheimer’s disease, was also shown in vitro and confirmed by the in vivo text on *Galleria mellonella*. Overall, the obtained results confirm once again the health potential of strawberries; however, the efficacy is dependent on high quality products submitted to correct processing flow charts.

## 1. Introduction

As is well known, many recent reports show that the onset and growth of metabolic syndrome, diabetes, inflammation, cancer, neurological and cardiovascular diseases are etiologically correlated to the reduced consumption of antioxidant molecules contained in fruits and greens. As a counterpart, the well-recognized health potential associated with the daily consumption of vegetables and fruits has represented the object of interest of many studies carried out during the last two decades [1].

Among other edible vegetables, berries, and more specifically strawberries, are particularly appreciated by this point of view, in relation to their high content of active interesting biomolecules. Belonging to the Rosaceae family, comprising a large number of edible species, *Fragaria vesca* L. and *Fragaria* × *ananassa* Duch. represent, respectively, the wild type and a species largely cultivated worldwide, both appreciated for their attractive characteristics, color, fragrance, taste, and high vitamin and mineral content.

These promising characteristics have prompted cultivators to trial many new crop crossings. Actually, many different species are cultivated, among which the Clery cultivar represented the interest of our previous [2] research. Many recent studies and reviews regarding the health benefits exerted by berries in general, and strawberries in particular, underline the special role of these fruits in the prevention of inflammation and cell damage [3,4]. It is well known that inflammation is dramatically correlated with cancer and chronic diseases [5] and that gut microbiota-derived products can induce inflammation and contribute to metabolic and degenerative illnesses [6].

The polyphenolic content and the health potential of *Fragaria* spp. were recently reported in a review [7], in which strawberries were evaluated as anticancer, anti-inflammatory, anti-obesity and chemoprotective agents, as well as for their potential in antimicrobial, anti-allergenic and anti-diabetic applications. Phenolic extracts of strawberry were also reported to inhibit *Candida albicans* growth [8].

Saxena et al. [9] recently reviewed the function of target enzymes and the role carried out by enzyme inhibitors in the etiology of Alzheimer’s disease (AD). Okello et al. [10] reported that cholinesterase inhibitors have been recently isolated from natural sources and tested for their health potential in the prevention of AD onset and progression. The involvement of apoptosis of cholinergic neurons of limbic and neocortical regions makes acetylcholine-mediated neurotransmission a key factor in contrasting the associated morbidity. In addition, the levels of butyrylcholinesterase (BChE) are increased in AD patients as compensation of a reduced expression of acetylcholinesterase (AChE) and the inhibition activity, exerted on both enzymes by flavan-3-ol compounds, gives these molecules renewed interest in the prevention strategies of the most seriously disabling neurodegenerative illness.

The tyrosinase enzyme, catalyzing the initial conversion of L-tyrosine in L-dopa and the subsequent oxidation in dopaquinone, which polymerizes producing melanin, is also involved in degenerative processes [11]. Implicated in the onset of the skin melanomas and in the increase of neuromelanin in nigral dopaminergic neurons, it was observed in higher contents in elderly humans and, moreover, in Parkinson’s patients [12]. Taslimi et al. [11] evaluated the anti-melanogenesis potential of different molecules obtained by natural sources, evidencing that natural phenols exerted significant inhibition activity towards tyrosinase enzyme (IC_50_ values ranging between 2.37 and 7.90 µM).

The anti-diabetic role exerted by foods containing functional ingredients has been known for many years and many recent papers deal with this activity. Different ellagitannins and polyphenols extracted by food and food by-products were recently evaluated in regards to their role in inhibiting α-amylase and α-glucosidase and so preventing obesity and associated comorbidities [13,14,15,16].

It is also well known that the human microbiota plays a preeminent role in maintaining the state of well-being. Moreover, in more recent years, papers have also suggested that the gut mycobiota may be deeply linked with health and disease. *Candida* spp. are common species in the mycobiota of healthy humans, but their colonization and overgrowth in the gastrointestinal tract has been considered to play a role in the pathogenesis of Crohn’s disease [17], in multiple sclerosis, in amyotrophic lateral sclerosis and AD [18]. Fungal materials and *Candida* DNA have been detected in cerebrospinal fluid and in different brain regions including external frontal cortex, cerebellar hemispheres, entorhinal cortex/hippocampus and choroid plexus of AD patients [19,20,21]. Moreover, it has been reported that *C. albicans* cells cause cerebritis and a mild memory impairment [22] and that a correlation exists between *Candida* infections, β-amyloid increase and AD severity [23].

As shown by the results obtained in our previous work [2], quality parameters of Clery strawberries, such as polyphenol content, are deeply influenced by applying different procedural techniques. The conclusions of this work allowed us to observe that anthocyanins, flavanols and phenolic acid content were strictly correlated to the applied workflow, highlighting protective effects exerted by pasteurization and negative effects caused by blanching treatment.

On the basis of this premise, the obtained homogenates from Clery strawberries, previously analyzed [2], and some new samples of a Clery graft on *Fragaria vesca* were submitted to a multimethodological evaluation of their bioactivity, aiming to predict their ability in preventing degenerative and chronic diseases, testing the enzymatic inhibition activity on acetyl (AChE) and butyrylcholinesterase (BChE), tyrosinase, α-amylase and α-glucosidase. Moreover, the antifungal activity was demonstrated in vitro against *C. albicans* planktonic and sessile cells and *C. albicans* biofilm formation, and in vivo in a *Galleria mellonella* waxworm infection model.

## 2. Results and Discussion

The results of our previous work [2] allowed us to discover how deeply the processing steps could affect the final characteristics of processed food, independently from their stated quality. Similar research on strawberries, strawberry juice and blackberries was carried out by Patras et al. [24] and Karacam et al. [25], and we have conducted analogue experiments on blueberries and goji berries [26,27]. Overall, these papers show that quality parameters are deeply influenced by thermal, microwave and homogenization treatments.

If different homogenization techniques alone are able to modify color and fragrance of starting food materials, when the homogenization is combined with very mild heat treatment, such as blanching, fast pasteurization or mild microwave treatment, important modifications in total phenolic content, total flavonoid content, antioxidant and antiradical activity are shown. Moreover, HPLC and GC analysis allows us to show significant differences in single molecule content as well as in several classes of analyzed compounds [2]. In Table 1 all the obtained and tested samples are reported.

With the aim to highlight the best workflow in terms of bioactives preservation, a further step was performed in the present work by selecting some of the obtained homogenates and monitoring them in relation to their residue bioactivity, evaluated in terms of enzyme inhibition and antifungal activity.

### 2.1. Enzyme Inhibition Activities

The effects of the strawberry samples on a large panel of important enzymes are reported in Table 2. Collectively, the inhibitory activities were homogeneous within the different treatments applied to the samples (blanching, homogenization, pasteurization, microwave-based effects), between defrosted or fresh fruits, and between Clery and Clery grafts on *Fragaria vesca*. Based on IC_50_ values (mg/mL), AChE inhibitory effects varied from 1.03 mg/mL (in WU) to 1.95 mg/mL (WU graft). Regarding BChE inhibition, the highest and lowest abilities were found in the PM graft (IC_50_: 1.81 mg/mL) and MP graft (IC_50_: 2.97 mg/mL). The tested samples exhibited similar tyrosinase ability and the range was found to be 1.60–2.03 mg/mL. WU and PM exhibited the strongest amylase inhibitory actions with those values of 3.76 and 8.83 mg/mL, respectively. Glucosidase inhibition abilities of the tested samples were close to each other (IC_50_: 1.00–1.44 mg/mL).

### 2.2. Antifungal Activity

The texted Clery strawberry samples showed in vitro antifungal activity against *C. albicans* strains, with GMMIC_50_ ranging from 19.36 µg/mL to 59.82 µg/mL and MIC_90_ values ranging from 68.13 µg/mL to 162.09 µg/mL. PU showed the best MIC_50_ values against ATCC 24433 with values of 2.92 µg/mL (Table 3).

Given the activity on planktonic cells, we wanted to see if activity on *Candida* biofilm could also be shown. In fact, *C. albicans* produces highly structured biofilms composed of yeast cells, pseudo-hyphal cells and hyphal cells, largely resistant to current antifungal drugs, representing a significant part of *C. albicans* virulence. The results obtained using different concentrations of PU extract against *Candida* biofilm formation were highly encouraging. These demonstrated that 80% of biofilm onset was inhibited with PU extract at a concentration of 500 µg/mL.

Moreover, just for an extract concentration of 62.5 µg/mL, about 60% of the biofilm generated by two of the three tested strains was inhibited (Figure 1).

The *G. mellonella* model, positively correlated with results from studies with *Caenorhabditis elegans* and mice [28], represents a simple and inexpensive alternative method for the rapid evaluation of antimicrobial drug effectiveness in vivo. On the basis of the encouraging results reported above, we proceeded by testing the PU extract activity on *C. albicans* ATCC10231, using the *G. mellonella* model. The obtained results (Table 4) confirmed the previously obtained results, showing that PU administered to the larvae is able to increase their survival, if compared to those only inoculated with *Candida*.

Flavonoids were demonstrated to inhibit fungal growth with different mechanisms, such as plasma membrane disruption, mitochondrial dysfunction, inhibition of RNA and protein synthesis, cell division, cell wall formation and the efflux pump systems [29]. As different flavonoids and flavanoids (with particular regard for epicatechin, which accounted for 316 μg/g in PU vs a median of 250 μg/g for the whole U Series) detected in strawberries were especially represented in sample PU [30], we could attribute the showed efficacy to this class of compounds. We can also conclude that Clery strawberry fruits, only mildly pasteurized and homogenized, could be a useful matrix to be investigated as innovative and cost-effective anti-mycotic agents.

### 2.3. Principal Component Analysis (PCA)

PCA (principal component analysis) was conducted in order to correlate the previously published data of fresh Clery strawberry samples [2] and the data related to fresh Clery strawberry graft samples (reported in Appendix A) with enzymatic inhibition activity and antifungal activity. The values were scaled using XLSTAT 2020 software, with the unit variance scale. As shown in Figure 2, related to PCA plots of the fresh analyzed Clery samples, the x-axis represents the first PCA dimension (F1) covering 51% of the total variance, whereas the y-axis is the second PCA dimension (F2) covering 29% of the total variance.

The red vectors indicate the investigated variables. It is possible to show that the main molecules correlated to enzymatic and antifungal activity are phenolic acids, flavonols and flavanols (the first quadrant in the PCA plot, Figure 2). From this point of view, the microwave-treated samples appear to show the best enzymatic inhibition activity, and the pasteurized samples are more correlated with the antifungal inhibition activity, suggesting a greater potential of samples submitted to this kind of treatment. In fact, as reported in Garzoli et al. [2], the microwave-treated and pasteurized samples showed the highest content of phenolic acids, flavanols and flavonols.

The pasteurized and only homogenized samples (fourth quadrant in the PCA plot, Figure 2), highly correlated with the anthocyanins content, do not show the highest enzymatic inhibition, indicating a lower involvement of the anthocyanins in this kind of activity, as also reported in another study available in the literature [31]. Finally, as expected, the blanching treatments (third quadrant in the PCA plot, Figure 2), having a greater impact on the strawberry phytocomplex [2], is correlated with the lowest enzymatic inhibition and antifungal activity.

In Figure 3 are reported the PCA plots of the fresh analyzed Clery graft samples. The x-axis represents the first PCA dimension (F1) covering 53% of the total variance, whereas the y-axis is the second PCA dimension (F2) covering 26% of the total variance.

In this case, the pasteurized and only homogenized samples showed the best enzymatic inhibition activity towards tyrosinase and AChE, and blanched were more correlated with BChE. This difference in respect to the previously discussed results of Clery samples could be in relation to an evidently different phytocomplex composition (HPLC-DAD analysis results are reported in the Appendix A). The high anthocyanins content found in the Clery samples (134–236 µg/g fresh weight) [2] is significantly reduced in the Clery graft ranging between 37 to 190 µg/g fresh weight. This low anthocyanin content is also associated with a reduced content of phenolic acids (for example, caffeic acid is completely lacking), flavanols and flavonols.

In this second sample series, the interesting molecules seemed more preserved by pasteurization followed by only homogenization (second quadrant in the PCA plot, Figure 3) rather than the microwave treatment. These overall data highlighted how different very similar botanical species could be and, consequently, how much the correlated health effects could be modified in terms of specificity and magnitude. Results confirmed that the bioactivity is highly correlated to the phenolic acids and flavanols content, as previously reported in the literature [32].

In conclusion, as shown by Figure 4, in which the two series were compared, the cultivar Clery showed higher bioactivity for microwaved samples (phenolic acid content correlated with tyrosinase inhibition) and pasteurized samples (flavanol content correlated with anti-*Candida* activity) In the Clery graft, in which all bioactive molecules are less represented, only homogenization represented the best treatment (phenolic acid content correlated with amylase and tyrosinase inhibition).

## 3. Materials and Methods

### 3.1. Materials

Clery and Clery graft on *Fragaria vesca* strawberries were purchased fully ripe by the farm “Fragole di Carchitti” (Palestrina, RM, 41°50′ N 12°54′ E). A part was immediately analyzed as fresh fruit and another part was frozen at −80 °C and stored at −18 °C until strawberries were defrosted and the processing was performed (def and def graft). A small quantity of immature (imm) Clery was also supplied. Bidistilled water, ethanol, 85% formic acid and acetonitrile RS for HPLC were purchased from Sigma (Milan, Italy).

### 3.2. Processing

The fully ripe fresh or thawed whole fruits were gently cleaned, if necessary, by impurities and stems and carefully washed and dried on paper. Then, they were evaluated by CIEL*a*b* color analysis according to and submitted to the different work-up and extraction procedures needed by the HPLC analysis, as previously described [2]. Color and HPLC data of Clery graft on *Fragaria vesca* strawberry homogenates and extracts are reported in Appendix A.

### 3.3. Determination of Enzyme Inhibitory Effects

Inhibitory effects of the so obtained homogenized and thermally treated samples, and extracted with hydroalcoholic mixture as previously described [2] were tested against different enzymes: tyrosinase (from mushroom, EC 1.14.18.1), amylase (from porcine pancreas, EC 3.2.1.1), glucosidase (from *Saccharomyces cerevisiae*, EC 3.2.1.20) and cholinesterases (AChE: from electric eel, Type-VI-S, EC 3.1.1.7; BChE: from horse serum, EC 3.1.1.8). Standard inhibitors were used as positive controls (galantamine for cholinesterases; kojic acid for tyrosinase; acarbose for amylase and glucosidase). To provide comparison with standard inhibitors, IC_50_ values (mg/mL) were also given (IC_50_ is extract concentration required for scavenging 50% of enzyme inhibitory assays). Experimental details are given in our previous paper [33].

### 3.4. Antifungal Susceptibility Testing

#### 3.4.1. Antifungal Susceptibility of *C. albicans*

Antifungal susceptibility of *C. albicans* was determined according to standardized methods for yeast using the broth microdilution method (CLSI M27-A3, 2008; CLSI, 2012). The strains *C. albicans* ATCC 10231, ATCC24433, ATCC90028, 3153A coming from the American Type Culture Collection (ATCC, Rockville, MD, USA) were grown on Sabouraud dextrose agar (Sigma Aldrich, St. Louis, MI, USA) at 35 °C for 24 h. The final concentration of the inoculum was 1.0 × 10^3^–1.5 × 10^3^ CFU/mL. The fruit extracts were dissolved previously in dimethyl sulfoxide at concentrations 100 times higher than the highest tested concentration. The extracts were then serially diluted 2-fold across the 96-well plates. The final concentrations ranged from 1.95 to 1000 µg/mL, for all the tested samples, in RPMI 1640 medium (Sigma-Aldrich, St. Louis, MI, USA). Three experiments were performed on duplicate, on separate dates, three times for each tested extract. The panels were incubated at 35 °C. After 24 h, the minimal inhibitory concentration (MIC) was determined and the results are expressed as median. MIC_50_ was the lowest concentration that caused a prominent decrease (≥50%) in visible growth. The MIC_90_ was defined as the lowest drug concentration that caused ≥90% growth inhibition compared with the drug-free control.

#### 3.4.2. In Vitro Activity of Compounds against *C. albicans* Biofilms

The anti-biofilm activity was evaluated as previously described [34]. Briefly, 4% aqueous crystal violet was added for 45 min and 100 µL aliquots were taken from each well for absorbance measurement (595 nm). At least two experiments were performed on two separate experiments for each compound tested five times.

#### 3.4.3. *G. mellonella* Survival Assay

*G. mellonella* killing assays were carried out as described previously [35]. Briefly, 60 larvae of 0.3 ± 0.03 g (10 for each group) were selected. Two groups were inoculated with 10 μL of PU (1000 μg/mL) with or without 2 × 10^6^ cells of *C. albicans* ATCC 10231. A group of larvae were only pierced, a group treated with sterile PBS and a group treated with *C. albicans*. The larvae were then incubated at 35 °C and monitored for 72 h and considered died when they did not respond to physical stimulation (a slight pressure with forceps). Each experiment was repeated at least three times and reported as percent survival rate.

### 3.5. Statistical Analysis

All analysis was performed in triplicate and the results are depicted as mean ± SD. To detect differences among samples, one-way ANOVA (with Tukey’s test) was performed with Xlstat (Isle of Anglesey, Pentraeth, UK) 2017 software (*p* < 0.05 was considered statistically significant) for determining differences in the tested samples.

## 4. Conclusions

Differently treated strawberry samples obtained by *cv.* Clery and a graft of this on *F. vesca*, evaluated for the best expressed activity on enzymatic inhibition of amylase, tyrosinase and acetylcholinesterase and for anti-*Candida* activity, confirmed the already reported potential of this edible fruit. Seasonal limited availability and consequent applied treatments aiming to preserve highly perishable foods, as previously shown, and herein confirmed, could deeply affect the phytochemical composition with high impact on the correlated bioactivity. Moreover, a great difference in terms of polyphenolic content was highlighted between the two analyzed series, despite the botanical parental filial relationship.

Clery strawberries and Clery graft were shown to be able to modulate the enzymatic activity of all the tested enzymes, with best results obtained on tyrosinase inhibition and a slight preference for BChE in respect to AChE. Overall, Clery graft samples are less active, in agreement with the lower content of bioactive molecules.

The Clery series, tested along with some selected thawed and immature samples, on *C. albicans* activity, showed a good in vitro inhibiting potential confirmed by biofilm growth inhibition and by the in vivo test on infected *G. mellonella*, whose survival reached 60% at 48 h in respect to 0% of the reference, to which the strawberry extract was not administered.

The PCA analysis also confirmed a higher activity for the Clery samples, highlighting that, if microwaves and pasteurization represent the best treatments for this series, the only homogenization process is correlated with the best expressed activity by the Clery graft series. This result, although the graft series appears less rich in phenolics and flavonoid compounds, could account for a minor impact on bioactive substances by the fruit degradation enzymatic processes, highlighting the possibility to apply a simpler, faster and more economic flowchart, which is a goal that deserves, in our opinion, further attention.

## Figures and Tables

**Figure 1 molecules-26-01731-f001:**
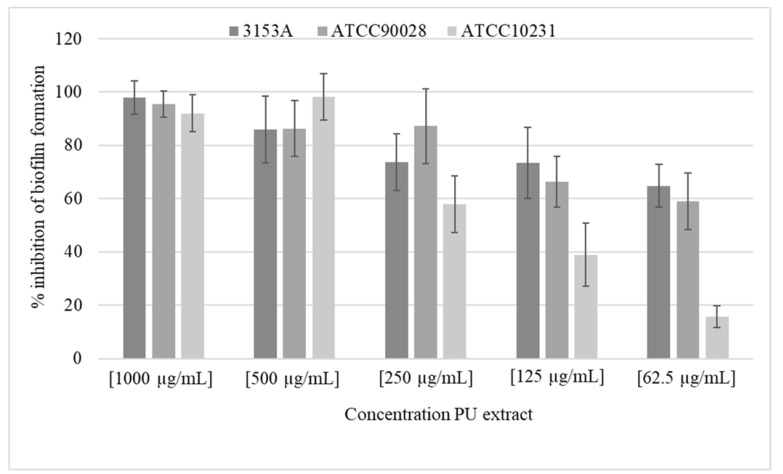
Inhibition of *C. albicans* biofilm formation using PU sample.

**Figure 2 molecules-26-01731-f002:**
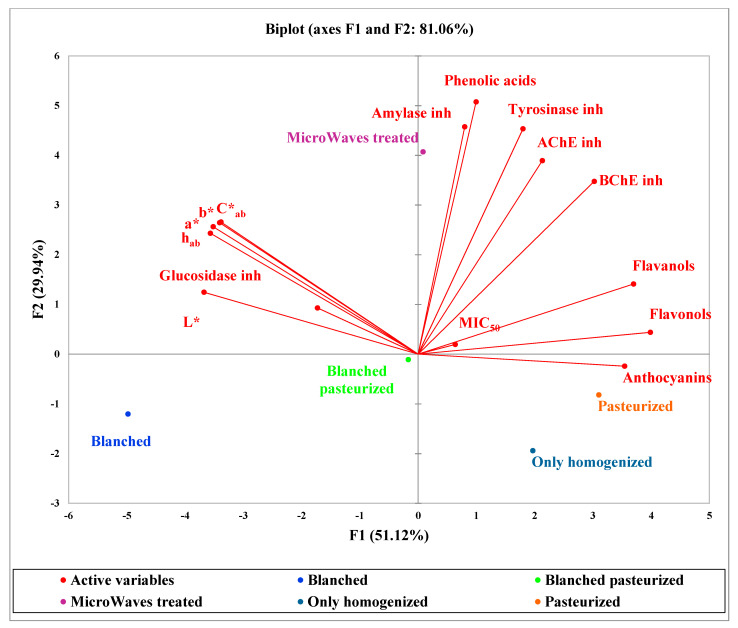
Principal component analysis (PCA) plots of the fresh analyzed Clery samples evaluated as series.

**Figure 3 molecules-26-01731-f003:**
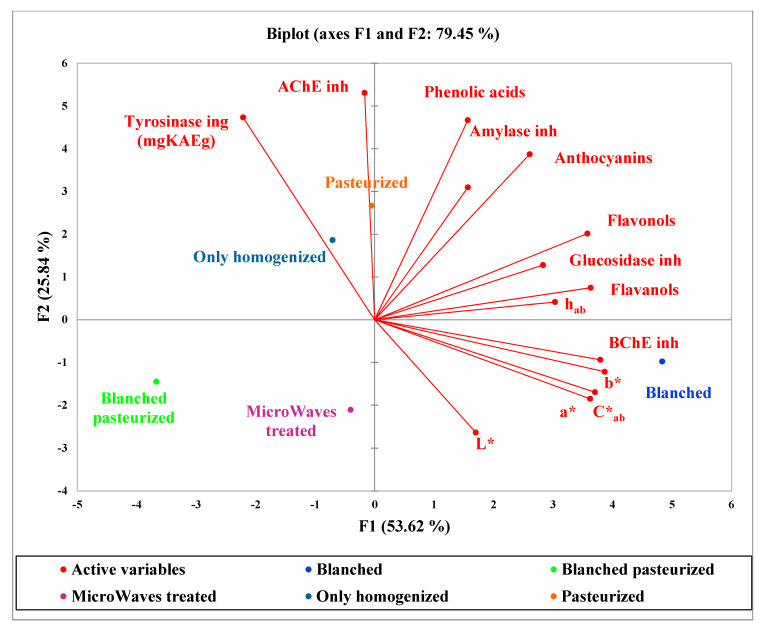
Principal component analysis (PCA) plots of the fresh analyzed Clery graft samples evaluated as series.

**Figure 4 molecules-26-01731-f004:**
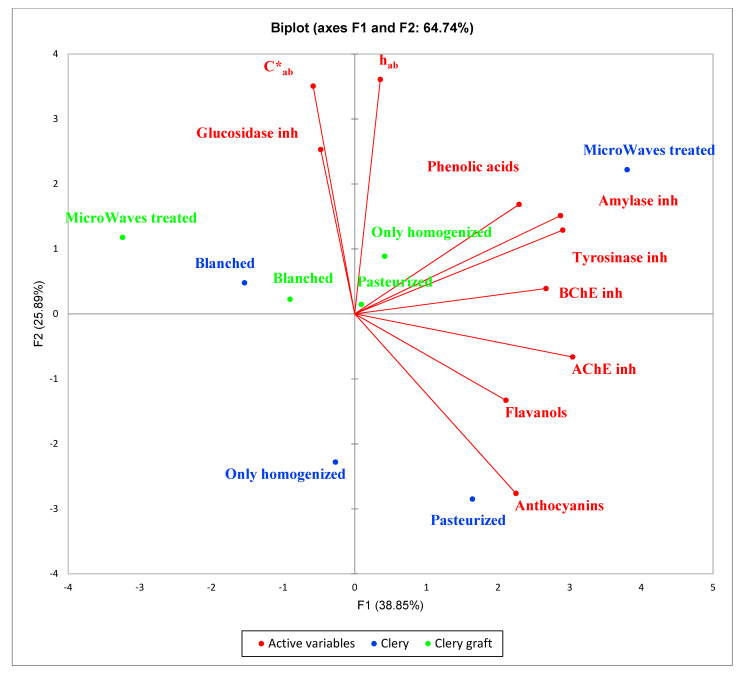
Principal component analysis (PCA) plots of the all strawberry samples analyzed.

**Table 1 molecules-26-01731-t001:** Sample legend.

Sample	Treatment	Sample	Treatment
M	Homogenized with mixer	U	Homogenized with Ultraturrax^®^
M def	Defrosted and homogenized with mixer	U def	Defrosted and homogenized with Ultraturrax^®^
MP	Homogenized with mixer and pasteurized	UP	Homogenized with Ultraturrax^®^ and pasteurized
BM	Blanched and homogenized with mixer	BU	Blanched and homogenized with Ultraturrax^®^
BMP	Blanched, homogenized with mixer and pasteurized	BUP	Blanched, homogenized with Ultraturrax^®^ and pasteurized
WM	Treated with microwaves and homogenized with mixer	WU	Treated with microwaves and homogenized with Ultraturrax^®^
PM	Pasteurized and homogenized with mixer	PU	Pasteurized and homogenized with Ultraturrax^®^
PM def	Defrosted, pasteurized and homogenized with mixer	PU def	Defrosted, pasteurized and homogenized with Ultraturrax^®^
PM imm.	Immature fruits pasteurized and homogenized with mixer	PU imm.	Immature fruits pasteurized and homogenized with Ultraturrax^®^
M graft	Graft Clery/*F. vesca* homogenized with mixer	U graft	Graft Clery/*F. vesca* homogenized with Ultraturrax^®^
M def graft	Graft Clery/*F. vesca* defrosted and homogenized with mixer	U def graft	Graft Clery/*F. vesca* defrosted and homogenized with Ultraturrax^®^
MP graft	Graft Clery/*F. vesca* homogenized with mixer and pasteurized	UP graft	Graft Clery/*F. vesca* homogenized with Ultraturrax^®^ and pasteurized
BM graft	Graft Clery/*F. vesca* defrosted and homogenized with mixer	BU graft	Graft Clery/*F. vesca* defrosted and homogenized with Ultraturrax^®^
WM graft	Graft Clery/*F. vesca* treated with microwaves and homogenized with mixer	WU graft	Graft Clery/*F. vesca* treated with microwaves and homogenized with Ultraturrax^®^
PM graft	Graft Clery/*F. vesca* pasteurized and homogenized with mixer	PU graft	Graft Clery/*F. vesca* pasteurized and homogenized with Ultraturrax^®^
BMP graft	Graft Clery/*F. vesca* defrosted and homogenized with mixer	BUP graft	Graft Clery/*F. vesca* defrosted and homogenized with Ultraturrax^®^

**Table 2 molecules-26-01731-t002:** Enzyme inhibitory results (IC_50_: mg/mL) of strawberry samples and standards against different enzymes.

Sample	AChE Inhibition	BChE Inhibition	Tyrosinase Inhibition	α-Amylase Inhibition	α-Glucosidase Inhibition
M	1.16 ± 0.01 ^jkl^	2.40 ± 0.33 ^ab^	1.92 ± 0.01 ^abcdefg^	12.84 ± 4.19 ^defg^	1.00 ± 0.01 ^d^
U	1.15 ± 0.01 ^kl^	2.23 ± 0.48 ^ab^	1.94 ± 0.01 ^abcde^	16.87 ± 2.28 ^abcde^	1.00 ± 0.01 ^d^
M def	1.40 ± 0.01 ^efg^	2.17 ± 0.27 ^ab^	1.79 ± 0.01 ^hijk^	17.02 ± 0.31 ^abcd^	1.00 ± 0.01 ^d^
U def	1.17 ± 0.01 ^jk^	2.17 ± 0.27 ^ab^	2.01 ± 0.02 ^a^	21.18 ± 0.13 ^a^	1.00 ± 0.01 ^d^
MP	1.23 ± 0.02 ^ijk^	2.37 ± 0.26 ^ab^	1.81 ± 0.05 ^ghijk^	10.65 ± 2.28 ^fg^	ni
UP	1.19 ± 0.01 ^jk^	2.27 ± 0.36 ^ab^	1.87 ± 0.04 ^cdefghi^	14.10 ± 2.66 ^cdef^	1.00 ± 0.01 ^d^
BM	1.25 ± 0.03 ^hij^	2.967 ± 0.43 ^a^	1.95 ± 0.02 ^abcd^	20.12 ± 0.17 ^ab^	1.00 ± 0.01 ^d^
BU	1.51 ± 0.05 ^d^	2.18 ± 0.07 ^ab^	1.97 ± 0.02 ^abc^	19.82 ± 0.81 ^ab^	1.01 ± 0.01 ^d^
BMP	1.21 ± 0.01 ^ijk^	2.13 ± 0.35 ^ab^	1.94 ± 0.09 ^abcde^	12.57 ± 0.92 ^defg^	1.00 ± 0.01 ^d^
BUP	1.25 ± 0.0 ^hij^	2.40 ± 0.38 ^ab^	1.83 ± 0.02 ^efghij^	15.54 ± 1.63 ^bcde^	ni
WM	1.07 ± 0.01 ^lm^	2.25 ± 0.45 ^ab^	1.92 ± 0.05 ^abcdefg^	12.10 ± 2.61 ^efg^	1.00 ± 0.01 ^d^
WU	1.03 ± 0.01 ^m^	1.99 ± 0.06 ^ab^	1.60 ± 0.02 ^l^	3.76 ± 0.05 ^h^	1.01 ± 0.01 ^d^
PM	1.16 ± 0.02 ^jkl^	2.06 ± 0.30 ^ab^	1.87 ± 0.02 ^cdefghi^	8.83 ± 2.09 ^g^	ni
PU	1.34 ± 0.02 ^gh^	2.07 ± 0.27 ^ab^	1.73 ± 0.02 ^jk^	14.25 ± 0.39 ^cdef^	1.00 ± 0.01 ^d^
M graft	1.14 ± 0.01 ^kl^	2.35 ± 0.20 ^ab^	1.77 ± 0.0 ^1ijk^	16.82 ± 0.56 ^abcde^	1.00 ± 0.01 ^d^
U graft	1.46 ± 0.01 ^def^	2.19 ± 0.32 ^ab^	1.86 ± 0.02 ^cdefghi^	18.32 ± 1.03 ^abc^	1.01 ± 0.01 ^d^
M def graft	1.66 ± 0.03 ^c^	2.62 ± 0.43 ^ab^	2.03 ± 0.05 ^a^	17.20 ± 0.57 ^abcd^	1.24 ± 0.08 ^b^
U def graft	1.37 ± 0.03 ^fg^	2.25 ± 0.05 ^ab^	2.00 ± 0.05 ^ab^	16.69 ± 2.79 ^abcde^	1.06 ± 0.01 ^d^
MP graft	1.16 ± 0.01 ^jkl^	2.97 ± 0.18 ^a^	1.71 ± 0.02 ^kl^	14.05 ± 1.46 ^cdef^	1.00 ± 0.01 ^d^
UP graft	1.25 ± 0.04 ^hij^	2.50 ± 0.33 ^ab^	1.81 ± 0.02 ^ghijk^	15.48 ± 0.57 ^bcde^	1.00 ± 0.01 ^d^
BM graft	1.66 ± 0.03 ^c^	2.15 ± 0.25 ^ab^	1.97 ± 0.05 ^abc^	16.13 ± 0.07 ^bcde^	1.00 ± 0.01 ^d^
BU graft	1.40 ± 0.05 ^efg^	1.99 ± 0.26 ^ab^	1.94 ± 0.03 ^abcdef^	16.30 ± 0.39 ^bcde^	1.00 ± 0.01 ^d^
BMP graft	1.31 ± 0.01 ^ghi^	2.61 ± 0.52 ^ab^	1.84 ± 0.03 ^defghij^	18.33 ± 0.33 ^abc^	1.00 ± 0.01 ^d^
BUP graft	1.79 ± 0.05 ^b^	2.22 ± 0.15 ^ab^	1.86 ± 0.02 ^cdefghi^	17.86 ± 1.48 ^abc^	1.44 ± 0.05 ^a^
WM graft	1.88 ± 0.02 ^ab^	2.60 ± 0.53 ^ab^	1.94 ± 0.09 ^abcde^	17.94 ± 1.54 ^abc^	1.01 ± 0.01 ^d^
WU graft	1.95 ± 0.05 ^a^	2.13 ± 0.53 ^ab^	1.89 ± 0.03 ^bcdefgh^	15.66 ± 0.23 ^bcde^	1.00 ± 0.01 ^d^
PM graft	1.21 ± 0.02 ^ijk^	1.81 ± 0.13 ^b^	1.82 ± 0.03 ^fghij^	15.81 ± 1.12 ^bcde^	1.02 ± 0.01 ^d^
PU graft	1.48 ± 0.04 ^de^	2.29 ± 0.10 ^ab^	1.81 ± 0.01 ^ghijk^	12.73 ± 0.34 ^defg^	1.14 ± 0.03 ^c^
Galantamine	0.003 ± 0.0001 ^n^	0.007 ± 0.0001 ^c^	nt	nt	nt
Kojic acid	nt	nt	0.08 ± 0.01 ^m^	nt	nt
Acarbose	nt	nt	nt	0.86 ± 0.01 ^h^	1.28 ± 0.04 ^b^

Values are reported as mean ± SD of three parallel experiments; ni: no inhibition; nt: not tested. Different letters (a–n) in same column indicate significant differences in the tested samples (*p* < 0.05).

**Table 3 molecules-26-01731-t003:** Antifungal activity of Clery strawberry samples against four strains of *Candida albicans* (Antifungal activity was determined according to Clinical and Laboratory Standards Institute guidelines (CLSI document M38-A2, 2008). Minimal inhibitory concentration (MIC) was determined. MIC_50_ and MIC_90_ = the lowest drug concentration that prevented 50% and 90% of growth with respect to the untreated control, respectively. The values shown are the median from three independent measurements).

Sample	*Candida albicans*			
	ATCC 90028	ATCC10231	ATCC24433	3153A
	Median MIC_50_ (μg/mL)			
M	312.5	7.8	11.7	156.25
U	93.75	23.42	5.85	62.5
MP	257.8	7.8	3.9	62.5
UP	93.75	7.8	3.9	62.5
BM	156.25	7.8	5.85	62.5
BU	187.5	11.7	3.9	93.75
BMP	93.75	11.7	7.8	62.5
BUP	46.87	3.9	3.9	62.5
WM	312.5	7.8	5.85	62.5
WU	281.25	7.8	7.8	93.75
PM	312.5	11.7	11.7	156.25
PU	281.25	5.85	2.92	62.5
PM def.	375	46.87	5.85	31.25
PU def.	187.5	46.87	9.75	62.5
PM imm.	375	23.42	4.85	31.25
PU imm.	187.5	62.5	15.6	93.75
	**Median MIC_90_ (μg/mL)**			
M	>1000	62.5	23.42	156.25
U	1000	78.12	15.6	62.5
MP	>1000	62.5	7.8	62.5
UP	750	62.5	7.8	62.5
BM	1000	187.5	7.8	62.5
BU	1000	93.75	23.42	93.75
BMP	1000	62.5	11.7	62.5
BUP	1000	93.75	7.8	62.5
WM	>1000	187.5	15.6	62.5
WU	1000	62.5	19.52	93.75
PM	>1000	62.5	23.42	281.25
PU	>1000	250	7.8	62.5
**PM def.**	>1000	93.75	7.8	31.25
**PU def.**	1000	187.5	15.6	93.75
**PM imm.**	>1000	62.5	7.8	31.25
**PU imm.**	1000	187.5	31.25	125

**Table 4 molecules-26-01731-t004:** % of survival rate of *G. mellonella* infected with *C. albicans* and treated with PU after 24, 48 and 72 h of incubation.

	% of Survival Rate		
	24 h	48 h	72 h
Control larvae	100%	100%	100%
Pierced larvae	100%	100%	100%
PBS	100%	60%	60%
PU 1000 µg/mL	100%	100%	100%
*C. albicans* ATCC 10231	40%	0	0
*C. albicans* ATCC 10231 + PU 1000 µg/mL	100%	60%	40%

PBS: Saline phosphate buffer.

## Data Availability

Not applicable.

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
