# Peer review of "Health Potential of Clery Strawberries: Enzymatic Inhibition and Anti-Candida Activity Evaluation"

_molecules, 2021, doi:10.3390/molecules26061731_

Round 1
Reviewer 1 Report
The work presented for review contains interesting research on healthy potential of Stawberries both belonging to cultivar Clery (Fragaria x ananassa Duchesne ex Weston) and to a graft obtained by crossing Clery and Fragaria vesca L. and is one of many works on the impact of natural products on the prevention of civilization and chronic diseases. In the presented work homogenates of fruits were evaluated in relation to their enzymatic inhibition activity towards acetylcholinesterase and butyrylcholinesterase, α-amylase, α-glucosidase and tyrosinase - the enzymes involved in the onset of such chronic diseases as: diabetes, rheumatoid arthritis and Alzheimer’s disease etc. The results of the work showed that all of tested enzymes were modulated by the tested samples. I find the subject and content of the article interesting and worth publishing in Molecules, especially in the section: Natural Products Chemistry. The correlation of the activity of extracts and other preparations from pre-digested fruit with the content of active compounds responsible for this effect is very important in this article. I have no substantive objections to the article. Editorial corrections: please insert dots instead of commas in decimal numbers (Fig. 2, Table 1S and 2S).
Author Response
Response to the reviewers
Reviewer 1.
The work presented for review contains interesting research on healthy potential of Stawberries both belonging to cultivar Clery (Fragaria x ananassa Duchesne ex Weston) and to a graft obtained by crossing Clery and Fragaria vesca L. and is one of many works on the impact of natural products on the prevention of civilization and chronic diseases. In the presented work homogenates of fruits were evaluated in relation to their enzymatic inhibition activity towards acetylcholinesterase and butyrylcholinesterase, α-amylase, α-glucosidase and tyrosinase - the enzymes involved in the onset of such chronic diseases as: diabetes, rheumatoid arthritis and Alzheimer’s disease etc. The results of the work showed that all of tested enzymes were modulated by the tested samples. I find the subject and content of the article interesting and worth publishing in Molecules, especially in the section: Natural Products Chemistry. The correlation of the activity of extracts and other preparations from pre-digested fruit with the content of active compounds responsible for this effect is very important in this article. I have no substantive objections to the article. Editorial corrections: please insert dots instead of commas in decimal numbers (Fig. 2, Table 1S and 2S).
We thank the reviewer for the positive judge on our paper. We have corrected the Figure and the Tables in the Supplementary as indicated.

Reviewer 2 Report
Thank you for submitting the manuscript “Healthy potential of Clery strawberries: enzymatic inhibition and anti-Candida activity evaluation” to Molecules.
In my opinion, the manuscript is well delineated and has interesting experimental results that justify its scientific publication. However, some points need to be improved.
- Introduction: in my opinion the authors could provide more information indicating why the most diverse processing methods can interfere with bioactive compounds and enzymes.
- From the information that appears in the introduction it is not possible to understand why the choice of Candida albicans. Why not use microorganisms that are considered food spoilers or the main cause of food poisoning?
- in the discussion it is necessary to use more citations from articles that evaluate fruit preservation processes.
Author Response
Response to the reviewer
Reviewer 2.
We thank for the kind revision and the precious suggestions
Thank you for submitting the manuscript “Healthy potential of Clery strawberries: enzymatic inhibition and anti-Candida activity evaluation” to Molecules. In my opinion, the manuscript is well delineated and has interesting experimental results that justify its scientific publication. However, some points need to be improved.
Introduction: in my opinion the authors could provide more information indicating why the most diverse processing methods can interfere with bioactive compounds and enzymes.
We have implemented the introduction and the discussion with regard to the different applied procedures. This was the object of our previous published papers, so we have here reported only the most relevant conclusions in terms of procedural effects on the bioactive components and consequently on the enzyme modulation. With this aim, lines 94-99 in introduction and lines 111-115 in Results and discussion were inserted.
From the information that appears in the introduction it is not possible to understand why the choice of Candida albicans. Why not use microorganisms that are considered food spoilers or the main cause of food poisoning?
Candida albicans was chosen for its implication in several chronic diseases. We evaluated the effects of this food matrix in terms of anti-Candida activity on the basis of this implication in human being and not relatively to the capacity of food to counteract microorganisms by which they could be spoiled. This part was already explained in the introduction in lines 83-93
in the discussion it is necessary to use more citations from articles that evaluate fruit preservation processes.
As referred above, “Results and Discussion” were modified accordingly in lines 111-114 and four references were added:
Patras et al. (2009). Impact of high pressure processing on total antioxidant activity, phenolic, ascorbic acid, anthocyanin content and colour of strawberry and blackberry purées. Innovative Food Science & Emerging Technologies, 10(3), 308-313.
Karacam, et al. (2015). Effect of high pressure homogenization (microfluidization) on the quality of Ottoman Strawberry (F. Ananassa) juice. LWT-Food Science and Technology, 64(2), 932-937.
Cesa et al. (2017). Evaluation of processing effects on anthocyanin content and colour modifications of blueberry (Vaccinium spp.) extracts: Comparison between HPLC-DAD and CIELAB analyses. Food chemistry, 232, 114-123.
Patsilinakos et al. (2018). Carotenoid content of Goji berries: CIELAB, HPLC-DAD analyses and quantitative correlation. Food chemistry, 268, 49-56.

Reviewer 3 Report
1. General comment
The authors prepared several types of strawberry homogenates and tested enzyme inhibitory and anticandidal potential of these samples. The homogenates were presumably in the liquid form and as such were added to the reaction mixtures or, after dilution with dimethyl sulfoxide, to cell suspensions in microtiter plates for antifungal activity determination. If so, the homogenate concentration in the final mixtures should be rather presented in the μl/ml than in the μg/ml format.
2. Comments to the enzyme inhibition studies
a) Data presented in Table 2 are misleading. They probably represent 5 enzymatic activities measured in reaction mixtures containing strawberry homogenate samples. There are not any positive controls containing all components but homogenate samples. If positive controls are lacking, it is not possible to estimate the level of enzyme inhibition. So that, in the table there are not data on enzyme inhibition but only on difference in enzymatic activity among samples containing different homogenates. Either the table heading is changed or data for positive controls are included.
b) More importantly, the interpretation of data obtained by the authors does not seem to be correct. The authors say that (cit.) "In detail, tyrosinase inhibition was the most valuable in terms of kojic acid equivalents, whereas the effects on cholinesterases and carbohydrate-metabolizing enzymes were limited." This does to seem to be true, since the alpha-amylase activities were in the 0.35 - 0.06 range, so that inhibition of this activity by the most active homogenate was very substantial (>80%), whereas the tyrosinase activities were in the 48.22 - 40.45 range, so that for the most active homogenates, the inhibition level was lower than 20%.
c) It is not clear what data were obtained in the case of samples indicated as "not active". If no inhibition of enzyme activity was observed, the activity measureforin this sample should be maximal, i.e. equal to that of the positive control.
d) For the methods of enzyme activity determination, the authors refer the reader to their previous paper (ref 29). However, in that paper, there are only methods for determination of tyrosinase and alpha-amylase activity, not for alpha-glucosidase, AChE and BChE.
e) In the introduction, the authors explain the reason for their choice of 5 enzymes for the studies on effect of strawberry homogenates on their activity. However, these explanations make sense only if the enzymes used in the studies were of mammalian (preferentially human) origin. Meanwhile, the tyrosinase used in this work was from mushroom and origin of alpha-glucosidase, AChE and BChE is not known.
3. Comments to the anticandidal activity studies
3.1. The authors claim that the final concentrations of strawberry supernatants in the samples tested were in the 1.95 to 1000 μg/mL range. If so, why for some samples tested for activity against C. albicans ATCC 90028 values 1500 μg/mL are provided? Should be just >1000 μg/mL.
3.2. In the CLSI-recommended protocol, two-fold serial dilutions of agents tested are prepared. In this method, the usual practice is presentation of the result as that of the three matching values not the average of three non-matching values.
3.3. One of the C. albicans strains used in this study was apparently resistant (MIC90 values 750 - >1000 μg/mL), while the 3 other were susceptible MIC90 values 2.9 - ∼100 - 200 μg/mL). The geometric mean calculation for 4 values of which one is very different from 3 other ones makes in my opinion little sense and leads to unnecessary averaging, not reflecting the real difference.
4. Galleria mellonella survival assay
4.1. I do not understand why the PU concentration is in the μg/mL format. For such experiment one would expected the μl/g body weight format.
4.2. PU concentration in toxicity determination experiment (larvae treated with PU only) was 1000 μg/mL, while in the antifungal in vivo activity determination (larvae infected with C. albicans and treated with PU) it was 1000 times higher (1 g/mL). By the way, I could hardly imagine how such a huge amount of PU was applied to the larvae.
Author Response
Response to the reviewers:
Reviewer 3.
We wish to thank this reviewer for his/her careful revision and the numerous and interesting issues he/she raised. We think the work has been improved on the basis of the in-depth topics he/she suggested. We made corrections in order to ask to all the highlighted shortcomings.
General comment
The authors prepared several types of strawberry homogenates and tested enzyme inhibitory and anticandidal potential of these samples. The homogenates were presumably in the liquid form and as such were added to the reaction mixtures or, after dilution with dimethyl sulfoxide, to cell suspensions in microtiter plates for antifungal activity determination. If so, the homogenate concentration in the final mixtures should be rather presented in the μl/ml than in the μg/ml format
The homogenates were evaluated after hydroalcoholic extraction as described in our previous paper. This aspect was not highlighted in the present paper and we agree with the reviewer that it was difficult to understand the applied procedure. Lines 259-262 were modified accordingly, in order to better explain the adopted methods.
- Comments to the enzyme inhibition studies
- a) Data presented in Table 2 are misleading. They probably represent 5 enzymatic activities measured in reaction mixtures containing strawberry homogenate samples. There are not any positive controls containing all components but homogenate samples. If positive controls are lacking, it is not possible to estimate the level of enzyme inhibition. So that, in the table there are not data on enzyme inhibition but only on difference in enzymatic activity among samples containing different homogenates. Either the table heading is changed or data for positive controls are included.
Thanks so much for this question. We have used positive controls for all enzyme inhibitory assays. The enzyme inhibitory activities of the extracts were evaluated as equivalents of standard inhibitors per gram of the homogenates (galantamine for acetylcholinesterase and butyrylcholinesterase, kojic acid for tyrosinase, and acarbose for α-amylase and α-glucosidase inhibition assays). The way we have presented the data is very common as evidenced by previous publications listed hereunder.
In this perspective, our results tend to align with previous peer reviewed published scientific literature and humbly request to accept this format (Sut et al. (2020) Industrial Crops and Products, 147, 112246; Sut et al. (2020). Food Research International, 129, 108877; Sinan, (2020) Antioxidants, 9(2), 163; Rocchetti et al (2019). Antioxidants, 8(12), 632; Zengin, G.., et al (2019), Food and Chemical Toxicology, 127, 237-250; Zengin G., et al., (2019), Industrial Crops and Products, 135, 107-121). In addition, we have been changed the table title.
- b) More importantly, the interpretation of data obtained by the authors does not seem to be correct. The authors say that (cit.) "In detail, tyrosinase inhibition was the most valuable in terms of kojic acid equivalents, whereas the effects on cholinesterases and carbohydrate-metabolizing enzymes were limited." This does to seem to be true, since the alpha-amylase activities were in the 0.35 - 0.06 range, so that inhibition of this activity by the most active homogenate was very substantial (>80%), whereas the tyrosinase activities were in the 48.22 - 40.45 range, so that for the most active homogenates, the inhibition level was lower than 20%.
We have improved this part (lines 129-139).
- c) It is not clear what data were obtained in the case of samples indicated as "not active". If no inhibition of enzyme activity was observed, the activity measureforin this sample should be maximal, i.e. equal to that of the positive control.
We have changed “not active” as “no inhibition”.
- d) For the methods of enzyme activity determination, the authors refer the reader to their previous paper (ref 29). However, in that paper, there are only methods for determination of tyrosinase and alpha-amylase activity, not for alpha-glucosidase, AChE and BChE.
We apologize for the mistake. We have replaced the reference.
- e) In the introduction, the authors explain the reason for their choice of 5 enzymes for the studies on effect of strawberry homogenates on their activity. However, these explanations make sense only if the enzymes used in the studies were of mammalian (preferentially human) origin. Meanwhile, the tyrosinase used in this work was from mushroom and origin of alpha-glucosidase, AChE and BChE is not known.
Thanks so much for your question. We have explained the selection of these enzymes in the introduction section. We used the enzymes from different sources. For example, tyrosinase is from mushroom; amylase from porcine pancreas; glucosidase from Saccharomyces cerevisiae, AChE is from electric eel and BChE is from equine serum. You could see the enzyme sources in the in the revised version. (lines 260-262)
The sources for enzymes have similar homology to human sources at high rate, therefore, these sources are preferred to determine the first insight of enzyme inhibitory potential of plant extracts or compounds. Orhan (2013) reported similar homology between electric ell and human AChEs. (Current Neupharmocology, 11, 379-387); Moorad et al (1999) and Dafferner et al (2017) explained similar homology between human and equine serum BChE (1999, Toxicology Methods, 9: 219-227; Dafferner et al (2017, Chemico-Biological Interactions, 266, 17-27), Brayer et al (1995, Protein Science, 4, 1730-1742) and Dessaux et al (2002, Biologia Bratislava, 57, 163-170) indicated similar three-dimensional structures of different sources of amylase Rost (1999, Protein Engineering, 12(2), 85), reported that structure characteristics in catalytic domain between yeast and human glucosidase were based on the identity level.
- Comments to the anticandidal activity studies
3.1. The authors claim that the final concentrations of strawberry supernatants in the samples tested were in the 1.95 to 1000 μg/mL range. If so, why for some samples tested for activity against C. albicans ATCC 90028 values 1500 μg/mL are provided? Should be just >1000 μg/mL.
All analyses were performed in three independent determinations in duplicate. The values shown are the median of MICs. To calculate this median when the value was >1000 we used the 2-fold value (2000). This explains the value of 1500. We understand that it can cause confusion so, following the reviewer's advice, we have now reported the data as > 1000.
3.2. In the CLSI-recommended protocol, two-fold serial dilutions of agents tested are prepared. In this method, the usual practice is presentation of the result as that of the three matching values not the average of three non-matching values.
The average was made on the basis of the results obtained from the same dilutions. The experiments were performed in duplicate and repeated three times. We have added this overall information in lines 275-280.
3.3. One of the C. albicans strains used in this study was apparently resistant (MIC90 values 750 - >1000 μg/mL), while the 3 other were susceptible MIC90 values 2.9 - ∼100 - 200 μg/mL). The geometric mean calculation for 4 values of which one is very different from 3 other ones makes in my opinion little sense and leads to unnecessary averaging, not reflecting the real difference.
We thank the reviewer for this correct consideration and therefore we eliminated the geometric mean.
- Galleria mellonellasurvival assay
4.1. I do not understand why the PU concentration is in the μg/mL format. For such experiment one would expected the μl/g body weight format.
We agree with the reviewer that methods were not explained in detail and therefore it could be difficult to understand. The adopted method was detailed by many authors (Lu, M., Yang, X., Yu, C., Gong, Y., Yuan, L., Hao, L., & Sun, S. (2019). Linezolid in combination with azoles induced synergistic effects against Candida albicans and protected Galleria mellonella against experimental candidiasis. Frontiers in microbiology, 9, 3142; Gu, W., Guo, D., Zhang, L., Xu, D., and Sun, S. (2016). The synergistic effect of azoles and fluoxetine against resistant Candida albicans strains is attributed to attenuating fungal virulence. Antimicrob. Agents Chemother. 60, 6179–6188. doi: 10.1128/AAC.03046-15; Gu, W., Yu, Q., Yu, C., and Sun, S. (2017). In vivo activity of fluconazole/tetracycline combinations in Galleria mellonella with resistant Candida albicans infection. J. Glob. Antimicrob. Resist. 13, 74–80. doi: 10.1016/j.jgar.2017.11.011). and a brief description was inserted in the text in lines 281-287.
4.2. PU concentration in toxicity determination experiment (larvae treated with PU only) was 1000 μg/mL, while in the antifungal in vivo activity determination (larvae infected with C. albicans and treated with PU) it was 1000 times higher (1 g/mL). By the way, I could hardly imagine how such a huge amount of PU was applied to the larvae.
Thanks to the reviewer for identifying a serious typo. A wrong value has been reported. The concentration used is obviously the same as the control without C. albicans, that is 1000 µg/ml. The value has been corrected.

Round 2
Reviewer 3 Report
The authors have properly addressed most of the comments, so that the ms. has been improved.
There is still one issue that needs further correction and this is a way of presentation of results of enzyme inhibition studies.
I cannot agree with the authors that this way of data presentation i.e. "equivalents of standard inhibitors per gram of the homogenates is very common as evidenced by previous publications listed hereunder". In my opinion, this is rather rare, may be exceptional, and probably limited to the papers listed by the authors of this ms: (Sut et al. (2020) Industrial Crops and Products, 147, 112246; Sut et al. (2020). Food Research International, 129, 108877; Sinan, (2020) Antioxidants, 9(2), 163; Rocchetti et al (2019). Antioxidants, 8(12), 632; Zengin, G.., et al (2019), Food and Chemical Toxicology, 127, 237-250; Zengin G., et al., (2019), Industrial Crops and Products, 135, 107-121).
All the papers cited above share at list one coauthor (Zenkin G. ), so that they come from the same group (although some of other coauthors are different). It seems therefore, that this way of presentation is an original idea of Dr. Zenkin and coworkers. If I am wrong, the authors should present other citations to papers of another groups applying the same approach. More importantly, in my opinion this way of presentation is not understandable, at least for the readers of journals like MOLECULES. As one of them, dealing with enzyme inhibition studies in my everyday research, I do not understand what enzyme inhibition in terms of "equivalents of standard inhibitors per gram of the homogenates" mean. Particularly, comparing values obtained for different samples tested against one of the enzymes, it is not clear (at least for me) if the higher value means that the inhibitory effect was stronger or lower. So that, if the authors insist on leaving unchanged their way of presentation (instead of changing them to really generally accepted ways (of which percent of inhibition of enzyme activity is the most obvious), they must provide a clear explanation of all these aspects, also in terms of positive controls in M&M section.
Author Response
The authors have properly addressed most of the comments, so that the ms. has been improved.
There is still one issue that needs further correction and this is a way of presentation of results of enzyme inhibition studies.
I cannot agree with the authors that this way of data presentation i.e. "equivalents of standard inhibitors per gram of the homogenates is very common as evidenced by previous publications listed hereunder". In my opinion, this is rather rare, may be exceptional, and probably limited to the papers listed by the authors of this ms: (Sut et al. (2020) Industrial Crops and Products, 147, 112246; Sut et al. (2020). Food Research International, 129, 108877; Sinan, (2020) Antioxidants, 9(2), 163; Rocchetti et al (2019). Antioxidants, 8(12), 632; Zengin, G.., et al (2019), Food and Chemical Toxicology, 127, 237-250; Zengin G., et al., (2019), Industrial Crops and Products, 135, 107-121).
All the papers cited above share at list one coauthor (Zenkin G. ), so that they come from the same group (although some of other coauthors are different). It seems therefore, that this way of presentation is an original idea of Dr. Zenkin and coworkers. If I am wrong, the authors should present other citations to papers of another groups applying the same approach. More importantly, in my opinion this way of presentation is not understandable, at least for the readers of journals like MOLECULES. As one of them, dealing with enzyme inhibition studies in my everyday research, I do not understand what enzyme inhibition in terms of "equivalents of standard inhibitors per gram of the homogenates" mean. Particularly, comparing values obtained for different samples tested against one of the enzymes, it is not clear (at least for me) if the higher value means that the inhibitory effect was stronger or lower. So that, if the authors insist on leaving unchanged their way of presentation (instead of changing them to really generally accepted ways (of which percent of inhibition of enzyme activity is the most obvious), they must provide a clear explanation of all these aspects, also in terms of positive controls in M&M section.
Response: Thanks so much for your attention and question. We can understand your concerns and the standard equivalent way is not common. However, because the unit of enzymes used in the literature is so different, this fact could not reflect a logical comparison. This is our team approach and we have published several papers with this experimental protocol. However, as kindly requested, we have changed the presentation for enzyme results and the IC50 values have been added in the revised version. Also, the text has been revised with these values. I hope that the revised version is now clear and easily to understand.